# Predicting the Intention to Sort Waste at Home in Rural Communities in Lebanon: An Application of the Theory of Planned Behaviour

**DOI:** 10.3390/ijerph19159383

**Published:** 2022-07-31

**Authors:** Marco Bardus, May A. Massoud

**Affiliations:** 1Institute of Applied Health Research, College of Medical and Dental Sciences, University of Birmingham, Birmingham B15 2TT, UK; m.bardus@bham.ac.uk or; 2Department of Health Promotion and Community Health, Faculty of Health Sciences, American University of Beirut, Beirut 1107 2020, Lebanon; 3Department of Environmental Health, Faculty of Health Sciences, American University of Beirut, Beirut 1107 2020, Lebanon

**Keywords:** solid waste management, waste sorting and recycling, psychological factors, theory of planned behaviour

## Abstract

Low- and middle-income countries (LMICs) such as Lebanon have limited technical, economic, and social infrastructures to manage municipal solid waste properly. Understanding what motivates citizens to sort waste at home is paramount to designing effective, efficient, and equitable waste management interventions. Within the solid waste management project “RES-Q” in Southern Lebanon, we investigated the socio-cognitive predictors of waste sorting in a sample of 767 households from the targeted area using the Theory of Planned Behaviour (TPB). Perceived behavioural control (β = 0.96, *p* < 0.001), perceived norms (β = −0.30, *p* < 0.001), and current behaviour (β = 0.06, *p* < 0.001) were the strongest predictors of intention; attitude toward separating waste was not a significant predictor (β = 0.04, *p* = 0.3881). Consequently, future behavioural interventions should build capability and opportunity to perform the behaviour before normalising it. For example, citizens should receive bins and bags to separate waste and be shown how to perform the behaviour and how easy and convenient it is to increase their behavioural control. In parallel, waste collection and treatment infrastructures must be in place so that citizens can see that sorting waste is a social norm. These actions will ensure the success of future behavioural interventions within the RES-Q project and beyond.

## 1. Introduction

The global yearly waste production reached 2.1 billion tons, a figure expected to reach 3.4 billion tons by 2050 [1]. While most of this waste is generated in high-income countries, the World Bank estimated that 129 million tons were developed in the Middle East and North Africa region [1], the majority of which is municipal solid waste, resulting from different household activities [2]. If improperly managed, solid waste has a detrimental effect on the environment. Examples of improper waste management solutions include open dumping and open burning of waste which result in soil, water, and air pollution; these practices, standard in many low- and middle-income countries (LMICs) [3], also have social and health impacts.

Globally, the management of solid waste entails significant economic and environmental challenges. Circular economy models are perceived as an alternative approach to materials management to traditional disposal-oriented linear economic systems [4,5]. The primary purpose of circular models is to mitigate the environmental stresses generated by resource-depleting anthropogenic activities by prolonging the lifecycle of products and extracted resources to the fullest while maximising production efficiency. The prime output of circular economy models centres around improving the circularity of resources to decouple economic growth from environmental deterioration [4,5,6,7].

For waste material to be efficiently and effectively reused or recycled, it should be separated at the household level to avert contamination, resulting in lower quality products [8]. Separating waste is an essential responsibility for citizens to advance waste recycling and the circular economy [8]. Generally, cities that implemented source segregation measures significantly reduced the waste that ended in landfills and increased recycling rates [9]. For example, in Sweden, between 1975 and 2016, landfilling decreased from 62% to less than 1%, and recycling increased from 6% to 35% [10].

Household waste segregation is challenging in developing countries [11], where it tends to be managed by municipalities, non-governmental organisations, or the private sector [12,13]. The informal sector plays a vital role in the waste value chain, especially in LMICs, where they recirculate and reprocess a significant proportion of the generated waste stream, thereby extending the lifecycle of certain products [14,15,16]. Their activities aid in reducing disposal rates, creating job opportunities, and diminishing formal waste operators’ costs. However, despite being an invaluable part of established solid waste structures, informal workers are often marginalised by governmental authorities in developing countries [17].

Solid waste mismanagement is a constant problem for Lebanon, a small LMIC in the Eastern Mediterranean region facing a waste management crisis since 2015 due to a lack of proper infrastructure and political instabilities [18,19]. Recent estimates show that approximately 2000 tons of municipal solid waste are produced annually, consisting of about 55% organic waste and 37% recyclable materials [20,21]; in addition, only 8% of recyclables are recycled [22]. Due to political instability, the longevity and productivity of currently applied models are constantly being tested by the country’s alternating social, environmental, and economic conditions. Additionally, the lack of cost recovery mechanisms has jeopardised the financial continuity of the sector and hindered the infrastructure’s expansion [23].

Despite several initiatives undertaken by governmental authorities in recent years to address this recurrent dilemma [24], there is a tendency to maintain a disposal-oriented management scheme. As in the case of most LMICs, little attention is provided to the establishment of implementation mechanisms that improve the process of sorting waste in the household. What is the citizens’ role in solid waste management in this challenging environment? A few cross-sectional investigations on current solid waste management practices revealed that the main determinants are the general population’s limited knowledge about the principles of waste management (e.g., reducing, reusing, and recycling), the limited capacity of institutions responsible for solid waste management (i.e., municipalities) [25], limited technological infrastructure, and policies at the national level [22]. The problems are even more accentuated in rural areas because of the distance from the main traffic arteries and the inadequate road structure. While some evidence suggests that individual citizens are willing to support solid waste management projects [26], municipalities are urged to join forces to provide unified collection and disposal systems, maximising the use of limited resources.

In response to the ongoing nationwide garbage crisis, a group of researchers from a consortium of universities, local community, and civil society organisations designed and implemented a project entitled: “Development of a Model Solid Waste Management Program for the Protection of the Sainiq River Basin in Southern Lebanon”, further referred to as the “RES-Q Project”. The European Delegation supports this ongoing project in Lebanon within the “PROMARE” programme (EuropeAid/151108/DD7ACT/LB). The goal of the RES-Q project is to build a “Circular Economy” model [27] by redesigning resource flows and increasing waste minimisation and recycling in a targeted area in Southern Lebanon, covering two governorates (Nabatiyeh and South Lebanon). A conceptual overview of the project has been published by Rashid et al. [28].

An essential component of the RES-Q project was the development of behaviour change and communication strategies to encourage recycling, reducing, and reusing waste at the household level. Based on the community-based social marketing (CBSM) framework [29,30], the RES-Q project team, led by researchers from the American University of Beirut, conducted a series of formative research activities, including a baseline survey (described below), interviews, and consultations with the citizens residing in the targeted area. The information collected was used to select the target behaviours, identify barriers and benefits, and develop a Solid Waste Management Masterplan and strategies to be implemented and evaluated in a more extensive campaign.

## 2. Aims and Objectives of the Study

This study aimed to investigate the socio-cognitive and behavioural predictors of the intention to sort waste at the source among citizens residing in the targeted area of the RES-Q project in Southern Lebanon. We focused on sorting waste at home, which was considered a project priority.

Specifically, this study aimed to: (1) assess the attitude, perceived norms, perceived behavioural control, and intention about sorting waste, and (2) determine which of the constructs predicted the intention to sort waste.

## 3. Literature Review and Theoretical Framework

The literature on recycling behaviour, including sorting at source, is vast, and many studies investigate this behaviour’s psychological factors or determinants. For example, a review by Concari et al. [31] showed a complex interplay between individual-level factors and the environment, including social norms, laws and legislation, and cultural factors. Another recent systematic review by Raghu and Rodrigues [32] identified 80 studies (most conducted in high-income countries) which measured 52 overall variables associated with different solid waste management practices; 34 comprised psychological factors (attitude was the central factor), 8 situational factors (opportunity was the most critical factor), 5 community factors, and 5 demographic factors (with age being the predominant factor).

Theory-driven, evidence-based frameworks founded on behavioural theories are also considered effective in influencing behaviours [33]. Other reviews focusing on behavioural strategies to promote household recycling suggest that interventions addressing attitude, social norms, and capability are the most effective, as well as creating current or past behaviour [32,34]. A 2013 meta-analysis of studies focusing on recycling behaviour showed that social and moral norms were the most reported variables influencing behaviour [35]. Varotto and Spagnolli found that social modelling and environmental alterations were the most effective techniques used in behavioural interventions to promote the recycling of waste [34]. Similarly, a recent review focusing on using behavioural nudges to incentivise household recycling identified social norms and reminders of norms as the main predictor of behaviour [36].

A behavioural theory that incorporates social norms, attitudes, and capabilities is the Theory of Planned Behaviour (TPB) [37,38,39]. The TPB is considered one of the most prominent behavioural theories in the context of pro-environmental behaviours [34], consistently assessed and used to design interventions [32,33,40]. The theory postulates that behaviour is influenced by its behavioural intention, which is influenced by three core constructs, the attitude toward the behaviour, perceived behavioural control (PBC), and perceived norm. In this study, we adopted the most recent version of the TPB, as conceptualised in the “reasoned action approach” by Ajzen and Fishbein in 2010 [41]. According to this framework, behavioural intention includes behavioural expectation (i.e., the likelihood of performing the behaviour) and willingness to perform the behaviour. Attitude toward the behaviour includes its instrumental and experiential evaluations. PBC consists of the perceived ability to perform the behaviour and the confidence to execute it. Perceived norm consists of the perception about what others do (descriptive norm) and the perception of other people approving the behaviour (injunctive norm) [41]. The core constructs predicting intention are influenced by behavioural, normative, and control beliefs that individuals develop throughout their lives. These beliefs are, in turn, influenced by “background factors” that Ajzen and Fishbein grouped into individual, social, and information categories, acknowledging that there are potentially endless factors that might affect an individual’s belief system [41]. Among the most common individual-level background factors, Ajzen and Fishbein mentioned personality traits, mood, emotions, values, general attitudes, and past behaviour; social factors encompass age, gender, income, and culture; information factors infer the knowledge and exposure to media or interventions that raise awareness about the behaviour [41].

The TPB has consistently shown its utility and efficacy in explaining recycling in European countries [42,43,44] and household solid waste segregation in various settings and contexts [45,46,47,48,49,50,51,52], including LMICs in the Eastern Mediterranean (Turkey and Greece [53,54,55]), Africa (South Africa [52] and Ghana [56]), and Asia (India [57], Iran [58], Saudi Arabia [59,60], and China [51]). According to Elhoushy and Lanzini’s systematic review [61], the core TPB constructs (i.e., attitude, perceived norm, and perceived behavioural control) were significantly related to behavioural intention in different contexts. For example, in a study focusing on food waste practices in Qatar [62], the authors found that attitude and subjective norms were positively and significantly associated with the behavioural intention to reduce waste, whereas PBC was negatively associated with it. PBC was also inversely related to food waste since the more control one exerted on reducing waste, the less food waste was produced [62]. Similar findings were reported in a TPB-inspired study conducted in Lebanon to assess the determinants of solid waste management practices in three different urban areas [25], demonstrating the critical role of attitudes toward the 3Rs and satisfaction with current practices. In a study done in Dammam, Saudi Arabia, household recycling was significantly predicted by social norms, and PBC was the strongest predictor of behavioural intention [59]. Similar findings were reported in a study based on a web-based survey diffused among Saudi Arabian citizens [60], with PBC being the strongest predictor of intention, followed by attitude and perceived norms.

## 4. Materials and Methods

This study uses data from a cross-sectional survey involving a representative sample of 767 households in the RES-Q project target area. The survey was based on an interviewer-administered questionnaire collected through tablets. The enumerators followed a specific protocol to complete the data collection procedure. The questionnaire, protocol, and procedures were approved by the American University of Beirut’s Institutional Review Board before the start of the study (ref. number: SBS-2019-0155).

### 4.1. Questionnaire

In this study, we adopted the most recent version of the TPB, as conceptualised in the “reasoned action approach” by Ajzen and Fishbein in 2010 [41]. We developed a contextualised bespoke questionnaire to assess the core TPB constructs (attitude, perceived norms, perceived behavioural control, and behavioural intention) following Ajzen’s recommendations [63]. Considering the cultural context of the RES-Q project, and to minimise respondent fatigue emerging from lengthy questionnaires, we used standard direct measures [63]. We developed an initial battery of questions through consultations with the target population. Then, we refined them, translated all questions into Arabic, and conducted a pilot testing between 7 and 12 June 2019, to ensure that the items were meaningful and internally consistent [63,64]. The questionnaire used is included in the Appendix A.

According to the “target, action, context, and time (TACT)” principle [41], the TPB questions were anchored to the same specific behavioural target and action “separating waste (e.g., organic from glass, plastic, cans, paper, etc.)”, with the context being in the household and the time “in the coming month”.

*Attitude toward separating waste* was assessed through three items matching the instrumental dimension (separating waste is “good”, “beneficial”, and “valuable”), and one item capturing the experiential dimension (separating waste is “enjoyable”). The items were measured through a 10-point scale, as in a semantic differential (harmful/beneficial; bad/good; worthless/valuable; and unenjoyable/enjoyable).

*Perceived norm* was assessed through two items: “Most people who are important to me approve that I separate waste” (injunctive norm); and “Most people like me are currently separating waste” (descriptive norm). The items were assessed through a 10-point Likert scale (strongly disagree/strongly agree).

*Perceived behavioural control* was assessed through two 10-point Likert-type items: “I am confident that I can separate waste”, linked to the dimension of autonomy or self-efficacy, and “It is totally up to me to separate waste”, related to the measurement of capacity or controllability.

The *behavioural intention* was assessed through two items, one 10-point likelihood scale (very unlikely/very likely) “In the coming month, how likely are you going to separate waste” (behavioural expectation); and one 10-point Likert scale: “In the coming month, I am planning to separate waste” (willingness).

Background information included current behaviour, socio-demographic, and other contextual factors. Current behaviour was operationalised through a binary question such as: “Do you separate waste?” followed by two questions, one assessing the type of recycled materials and another assessing the reasons for segregating and not segregating waste. Sociodemographic factors included respondents’ age (years), gender, education, and marital status.

Socioeconomic status of the household, a common source of variability in environmental studies [65], was operationalised using the crowding index [66], calculated as the number of people living in the household divided by the number of rooms, excluding the kitchen and bathrooms. For descriptive purposes, a categorical variable was created to determine the level of overcrowding: acceptable (less than one person per room), medium crowding (1–2 people/room), overcrowding (more than two people/room) [66].

Other contextual variables included satisfaction with the current waste collection system and treatment, measured through two 10-point scales (1 Extremely dissatisfied–10 Extremely satisfied). The general attitude toward reducing, reusing, recycling, and recovering waste [67] is as follows: “How important is for you to… reduce, that avoids creating waste; reuse, that is using a product again or for another purpose; recycle, using materials again; and recover, that is producing energy from waste”. All items were measured using 10-point scales (1 Not important at all–10 Essential).

### 4.2. Sampling and Procedures

Participants were citizens residing in the villages of the RES-Q project target area (see Figure 1a below). The area covered three Union of Municipalities: Iqlim el Tuffah (Nabatiyeh Governorate, ten towns, further referred to as “Region 1”), Jezzine (South Lebanon Governorate, Jezzine district, twenty-seven towns, “Region 2”), and Jabal el Rihan (South Lebanon Governorate, Jezzine district, six towns, and one village, “Region 3”). The total estimated number of households was 18,331, with 10,242 households (56%) representing Region 1, 6361 (35%) Region 2, and 1728 (9%) Region 3.

Based on the total number of households in the targeted area, assuming a confidence level of 99%, a 5% margin of error, and a maximum probability of 50% in sorting at source, we needed to collect data from at least 643 households. We used a random walk quota sample [68] stratified by the village to build a representative sample of the people in the area. We generated a map of the region based on the dataset provided by the United Nations Office for the Coordination of Humanitarian Affairs (OCHA) [69]. We used ArcGIS [70] to generate a series of random points for each village; the weight for the random number generator was the estimated number of households in each town. The resulting layer of randomly generated points with coordinates was exported to Google Maps (Figure 1b,c) and imported into Kobotoolbox [71], the software used to collect data on tablets.

After obtaining permission from the municipalities and security forces, enumerators approached households following a specific protocol. The enumerators used an invitation script and obtained oral consent before starting the interview. To participate in the study, respondents had to be adults, live in the target area, be responsible for decisions regarding household solid waste management and disposal and provide oral consent for participating in the survey. Respondents could drop out at any time without any consequence. To maximise variability, interviews were conducted on weekends (Friday, Saturday, and Sunday) and weekdays (Wednesday) and in two consecutive waves (14–26 June and 5–18 July 2019). The targeted area is presented in Figure 1a below. Figure 1b,c present the distribution of the data collected through Kobotoolbox.

### 4.3. Data Analyses

The response datasets were downloaded from Kobotoolbox and imported into Excel to be prepared for the analyses conducted with JASP [72]. Internal consistency of multi-item TPB constructs was evaluated using Cronbach’s alphas, interpreted as excellent (≥0.90), good (0.80–0.89), acceptable (0.70–0.79), questionable (0.60–0.69), poor (0.50–0.59), and unacceptable (<0.50) [73,74].

Confirmatory factor analysis (CFA) was used to test the measurement structure, and factorability of the TPB constructs [75]. Once acceptable internal consistency and factorability were assured, a structural equation model was used to test a TPB model, with attitude toward separating waste, perceived norm, and PBC predicting behavioural intention, controlling for current behaviour, sociodemographic factors (age, gender, education, and crowding index) and other psychological factors mentioned in Section 4.1.

We used several indices to evaluate the model’s goodness of fit with the data. Indices included the comparative fit index (CFI), the Tucker–Lewis Index (TLI), the root mean square error approximation (RMSEA), and the standardised root mean square residual (SRMR). An acceptable fit was assumed when CFI and TLI indices were above 0.90 and RMSEA and SRMR were below 0.08. An excellent fitting model was considered when CFI and TLI values were above 0.95 and RMSEA and SRMR below 0.05 [76].

## 5. Results

### 5.1. Sample Characteristics

The interviewers approached 769 households, and 767 agreed to participate (99.7% participation rate). The respondents were proportionally distributed across the area: Region 1 (439/767, 57%), Region 2 (268/767, 35%), and Region 3 (60/767, 8%). Table 1 below provides a summary of the participants’ sociodemographic profiles. Briefly, most respondents were female (58.5%), with a mean age of 46 years (SD = 14.6, range: 18–95), with a university-level education (28%) and secondary-level education (22%). Most respondents were married (72%). The household-level variable to express socioeconomic status was the crowding index, which was 1.1 (SD = 0.6, range: 0–8), with most of the sample (53%) being classified in the moderate crowding category (i.e., 1–2 people/room). In general, the respondents from Region 2 had higher socioeconomic status than Region 1 and Region 3, as indicated by the level of education and crowding index. We retained the latter as indicator of socioeconomic status in the structural equation model.

### 5.2. Attitudes toward Recycling, Reducing, and Reusin and Reasons for Separating Waste

As for the general attitudes toward the existing waste management system and behaviour on average, respondents from Regions 2 and 3 had very positive attitudes toward reducing, recycling, recovering, and reusing (all means were above 8.3 out of 10). The attitude toward reducing was highest among respondents from Regions 2 and 3. Respondents from Region 2 had significantly higher attitudes toward reducing, reusing, recycling, and recovering than those from Region 1.

Respondents were moderately satisfied with how the waste was collected and treated (average 6 over 10). The waste collection and treatment satisfaction scores were significantly higher in Region 2 than in Region 3 and Region 1.

Among those who reported separating at source (*n* = 196), the majority stated they separated plastic (130/196, 66%), glass (80/196, 41%), paper (79/196, 40%), aluminium, and other materials such as iron (64/196, 33%). The most frequent reasons for separating at source included environmental concerns: “recycling reduces landfills” (117/196, 60%), it “helps our climate problems” (76/196, 39%), and it “preserves resources and protects wildlife” (64/196, 33%); one aspect related to the value for money: “is good for the economy” (69/196, 35%).

Among those who did not sort at source (*n* = 556), the most frequent reasons were: “nobody is doing it in the village” (201/556, 36%), which is linked to the perceived norm, and “there is no system to collect recyclables” (180/556, 32%), which is related to the capability and actual behavioural control. Other reasons included the lack of time to bring to a drop-off point and to separate the garbage, which can be linked to the TPB factors of attitude toward the behaviour and the control over the behaviour (82/556, 15%).

### 5.3. TPB Indicators

As reported at the bottom of Table 1, on average, the sample demonstrated a relatively high attitude toward sorting at source (M = 8.6 on a 10-point scale), followed by behavioural intention (M = 7.2) and PBC (M = 6.7). The perceived norm displayed a moderate-to-low level (M = 4.9). There were significant differences across the three Regions in attitude (*p* = 0.003), perceived norms (*p* = 0.001), and intention (*p* = 0.020), but not PBC (*p* = 0.379). Respondents from Regions 2 and 3 showed higher scores than Region 1.

As for current behaviour, most respondents stated they did not separate waste (565/767, 74%). Respondents from Region 2 were more likely to separate waste compared to their counterparts living in Regions 1 and 3 (Χ^2^ = 23.260, df = 2, *p* < 0.001).

### 5.4. TPB Measurement Model

Table 2 displays the average internal consistency estimates, factor loadings, and respective residual variances about the items representing the core TPB constructs. Within each factor, all items showed excellent internal consistency (Cronbach’s alpha indices above 0.82).

A confirmatory factor analysis with the four TPB factors demonstrated an overall good fit with the data, as most of the indices showed good fit (Χ^2^ = 213.474, df = 29, *p* < 0.001; CFI = 0.970; TLI = 0.953; SRMR = 0.041; GFI = 0.995), except the RMSEA which showed poor fit (RMSEA = 0.091; 90% CI: 0.080–0.103, *p* < 0.001). Inspection of the modification index showed potential covariances between two items of the attitude factor (beneficial-good). After adding a correlation between these items, the RMSEA improved to reach an acceptable level (RMSEA = 0.078; 90% CI: 0.066–0.090, *p* < 0.001) and the other fit indices improved, showing good fit (Χ^2^ = 158.038, df = 28, *p* < 0.001; CFI = 0.979; TLI = 0.966; SRMR = 0.040; GFI = 0.997). Modification indices did not reveal other admissible covariances. These findings suggested that a structural equation model could fit the data.

### 5.5. TPB Structural Model

A structural TPB model was based on a maximum likelihood estimator, with robust error calculation and full-information maximum likelihood estimation of missing data for continuous variables. The model included all TPB constructs derived from the previous CFA model, with the fixed covariances in the attitude factor. The original TPB model achieved an overall good fit with the data (Χ^2^ = 156.559, df = 27, *p* < 0.001; CFI = 0.979; TLI = 0.965; SRMR = 0.040; GFI = 0.997; RMSEA = 0.079, 90% CI: 0.067–0.091, *p* < 0.001). The model explained 91.4% of the variance in behavioural intention. Inspection of the modification index showed no plausible item covariances to be added.

A model with current behaviour predicting the TPB latent factors fitted the data well (Χ^2^ = 194.627, df = 33, *p* < 0.001; CFI = 0.974; TLI = 0.957; SRMR = 0.040; GFI = 0.996; RMSEA = 0.080, 90% CI: 0.069–0.091, *p* < 0.001). Current behaviour explained an additional 3% of the variance in intention (reaching 94%), 20% of the variance in the perceived norm, 11% of the variance in PBC, and only 0.4% of the variance in attitude.

Finally, we tested a complete TPB model including current behaviour and gender, crowding index, general attitudes toward reducing, reusing, recycling, and recovering, and satisfaction with waste collection and treatment. The model achieved an overall good fit with the data (Χ^2^ = 310.042, df = 81, *p* < 0.001; CFI = 0.968; TLI = 0.946; SRMR = 0.028; GFI = 0.996; RMSEA = 0.061, 90% CI: 0.054–0.068, *p* < 0.001). Inspection of the modification indices showed another potential covariance between two attitude items (beneficial-valuable). After adding this covariance, the model improved slightly (Χ^2^ = 297.235, df = 80, *p* < 0.001; CFI = 0.969; TLI = 0.948; SRMR = 0.027; GFI = 0.996; RMSEA = 0.060, 90% CI: 0.052–0.067, *p* < 0.001) and no other plausible covariances appeared. The final structural equation model is displayed in Figure 2 below.

The model explained 94% of the variance in behavioural intention, 47% in attitude, 34% in PBC, and 33% in the perceived norm. Perceived norm was negatively correlated with attitude (r = −0.17, *p* < 0.001) and positively correlated with PBC (r = 0.29, *p* < 0.001); PBC was positively correlated with attitude (r = 0.13, *p* = 0.009).

PBC was the strongest direct predictor of intention (β = 0.96, *p* < 0.001), followed by perceived norm (β = −0.30, *p* < 0.001), whereas attitude was not significant (β = 0.04, *p* = 0.332). The behavioural intention was also directly predicted by current behaviour (β = 0.06, *p* = 0.042). Holding all factors constant, a one-unit increase on the PBC scale would have increased the behavioural intention scale by 1.02 units.

Significant sociodemographic factors predicting the core TPB variables included gender, which was marginally related to attitude (β = 0.06, *p* = 0.048), and the crowding index, which was negatively associated with attitude (β = −0.10, *p* < 0.001) and positively associated with the perceived norm (β = 0.12, *p* = 0.002). The attitudes toward reducing, reusing, recycling, and recovering showed significant associations with the core TPB constructs, except for the attitude toward recycling. The general attitude toward reducing waste was positively related to attitude (β = 0.22, *p* < 0.001), PBC (β = 0.10, *p* = 0.042), and perceived norm (β = 0.16, *p* = 0.001). The general attitude toward reusing waste was negatively related to the perceived norm (β = −0.21, *p* = 0.002) and positively associated with attitude (β = 0.13, *p* = 0.033). The prevailing attitude toward recovering waste was negatively related to the perceived norm (β = −0.11, *p* = 0.045) and positively associated with attitude (β = 0.36, *p* < 0.001) and PBC (β = 0.24, *p* < 0.001). The satisfaction with the current waste collection system was negatively related to PBC (β = −043, *p* < 0.001). Satisfaction with current waste treatment was negatively associated with attitude (β = −0.20, *p* < 0.001) and positively related to PBC (β = 0.56, *p* < 0.001). All regression coefficients are included in Appendix B.

## 6. Discussion

This is the first TPB-based study conducted in Lebanon to explain the intention to sort waste at home among a representative sample of households from rural areas. This study also aimed to determine which constructs of the TPB model predicted the intention to sort waste. Compared to the existing TPB literature, the data support the utility of the TPB in explaining behavioural intention. The tested TPB model explained 94% of the variance in behavioural intention, 46% in attitude, 34% in PBC, and 33% in the perceived norm. The high level of variance explained in behavioural intention is due to the inclusion of current behaviour and other background factors such as gender, crowding index, the generic attitudes toward recycling, reusing, reducing, and recovering, and satisfaction with the current waste management system (collection and treatment). While differences exist in the relative size and the direction of the effects, relatively high proportions of variance explained in intention were reported in other TPB-based studies that included other variables outside the core TPB constructs. For example, a study based on a sample of 827 adults recruited via Mechanical Turk (mTurk) reported a TPB model predicting 89% of the variance in behavioural intention [77]; another study targeting 400 households in Turkey reported a TPB model explaining 80% of the variance [53], and a study done in Greece, among 357 households in Greater Athens, reported 79% of the variance [55].

The sample demonstrated relatively high attitudes toward sorting at source, reducing, recycling, recovering, and reusing waste. The behavioural intention was moderately high as well as PBC, whereas the perceived norm displayed a moderate-to-low level. The fact that the attitudes and behavioural intention were high in the three areas is encouraging, as it indicates that awareness campaigns stressing the importance of sorting at source could evoke a positive response in the citizens. There were significant differences across the Regions due to different existing waste management systems. For instance, Regions 2 and 3 had organised past awareness campaigns promoting household waste sorting and set up a waste collection system, which was absent in Region 1. This can explain the differential levels of attitude, intentions, and perceived norms. PBC was not significantly different across the Regions, demonstrating that the skills and abilities of the citizens were similar. The low level of social norms can be explained by low behavioural compliance since only about 26% of the households declared they separated waste at home. The low behavioural compliance is consistent with reports and studies conducted in Lebanon [20,22,23,25,26].

However, attitude was not a strong predictor of behavioural intention, with perceived behavioural control (PBC), perceived norms, and current behaviour being the most robust predictors. The critical role of PBC in influencing behavioural intention was reported in other TPB studies targeting households in Turkey [53], Greece [54], Saudi Arabia [60], and India [57]. Social norm was a significant predictor of intention in TPB-based studies conducted in Saudi Arabia [52,59,60] and Qatar [62]. A non-significant relationship between attitude toward recycling and intention was reported in other TPB-based studies [53,77]. These differences across countries may be due to the different instruments used or, most likely, to different samples. Since there are no standardised and validated scales to assess the TPB constructs, each questionnaire is built differently. The number of items used to determine a latent factor also changes, introducing relevant variability. However, this aligns with Ajzen’s TPB questionnaire development recommendations [63]. A bespoke questionnaire meets the needs and characteristics of the target population, increasing the precision of the estimates and their validity [63].

Significant background factors influencing TPB latent constructs included current behaviour and the attitudes toward reducing, reusing, and recovering waste. The behavioural intention was indirectly and directly predicted by current behaviour. Past or present behaviour was found to significantly influence behavioural intention in studies conducted in various parts of the world, including the studies mentioned above conducted in Iran [58], Greece [55], and China [51]. The attitude toward sorting at source was positively influenced by the general attitudes toward reducing, reusing, and recovering waste but negatively associated with the crowding index. The perceived norms factor was positively related to the attitude toward reducing waste and crowding index but negatively associated with the attitudes toward reducing and recovering waste. PBC was also significantly related to the attitude toward reducing and salvaging waste. The role of these background factors supports the idea that the TPB constructs mediate the relationship between background factors and behavioural intention and future behaviour [41]. As there are potentially infinite background factors to control for [41], one can assume to account for those that are meaningful, based on literature evidence. In this case, the evidence suggested accounting for attitudes toward reducing, recycling, and reusing waste [67] and the household crowding index [66], which turned out to be significantly related to the outcome.

How can these findings be used for developing a behaviour change strategy in the context of the RES-Q project? First, interventions should focus on building behavioural skills linked to the PBC construct and critical mass to encourage the behaviour. This is because citizens already hold positive attitude toward the behaviour, but they may not know how to enforce it because there is no appropriate segregation, collection, and treatment system. We should start creating initiatives to build the capability and the opportunity to perform the behaviour, which is part of the PBC construct [78]. For example, citizens need to receive bins and bags that would allow them to separate waste. They would need to be trained to properly separate waste to build their capability or controllability of the behaviour. In addition, citizens need to feel confident about performing the behaviour and believe they can do it, having reasons to continue doing so. Interventions need to show citizens how to perform the behaviour and how easy and convenient it is, which addresses self-efficacy, following the evidence from many social marketing interventions [33,79]. In parallel, waste collection and treatment infrastructures must be in place so that citizens can see that sorting waste is the norm. The “social norms approach”, which has been shown effective in behavioural interventions using nudging techniques [36], can be effective only once a system is in place and people are using it.

Based on these results, we developed a pilot intervention to be implemented in a village that showed the least behavioural compliance and had just introduced a waste collection and treatment system. The pilot intervention was designed through a collaboration between the municipalities, a group of graduate students, and local non-governmental organisations as part of a service-learning-based “Social marketing for public health” course offered by the Faculty of Health Sciences [24,80]. The intervention is aimed to build the capacity of how to segregate and how to sort at the source. In addition, it also aims to encourage and empower people to initiate or continue systematic waste sorting. Thus, the campaign seeks to increase the percentage of people who believe they can reduce their waste and those who intend to adopt the desired behaviour. The intervention entails the distribution of sorting bags and video tutorials to demonstrate how to separate waste and heighten the target audience’s perceived behavioural control. The intervention will also have magnetic flyers to reinforce the video’s messages and reminders to perform the behaviour. The collection system will standardise the behaviour, thus targeting the subjective norms. This way, the residents will acknowledge that everyone is involved in the behaviour and that this is the norm. If the campaign is proven effective, it could be scaled up to target the entire village or other villages included in the RES-Q project.

## 7. Conclusions

This study demonstrates that it is vital to establish a consistent and efficient solid waste management system to enable sorting waste at home. Since perceived behavioural control and perceived norm were the strongest predictors of the intention to sort waste at home, future interventions should be focused on leveraging these TPB-based constructs to have a positive impact. To implement a successful intervention to promote sorting at the source in the targeted areas, it is necessary to build the citizens’ capacities and capabilities to perform the behaviour in their homes. At the same time, it is essential to have a solid waste management infrastructure capable of effectively and efficiently collecting the waste from households.

Citizens need to be empowered to sort waste at home by receiving bins and bags that would allow them to separate waste. Behaviour change communication campaigns can boost the citizens’ confidence and capability through instructional video tutorials and motivational messaging highlighting the ease of use, simplicity, and value of sorting at source, encouraging their perceived control. Once more citizens adopt the behaviour, awareness campaigns can be implemented to demonstrate the impact of the collective effort, building the idea that the behaviour is followed by many, hence using a social norm approach. Citizens need to believe that their actions are worthwhile, so transparent communication should demonstrate what happens to the waste through the collection system.

This study demonstrates how to use data to develop behaviour change strategies to promote sorting waste at the source in the areas targeted by the RES-Q project. However, the process followed can be adapted to be implemented in similar initiatives in Lebanon and other parts of the world.

## Figures and Tables

**Figure 1 ijerph-19-09383-f001:**
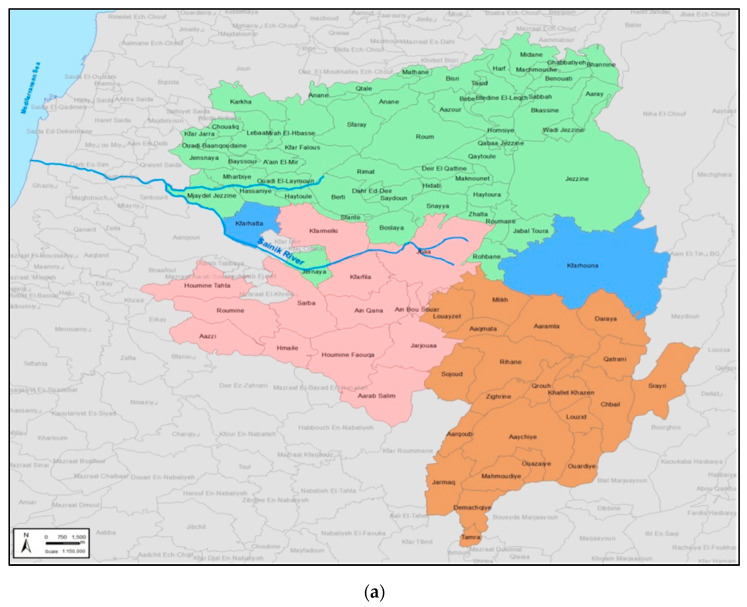
(**a**) The RES-Q project area (Region 1 is in red; Region 2 in green; and Region 3 in brown); (**b**) Kobotoolbox-generated map displaying the first wave of data collected (*n* = 380); and (**c**) map displaying the second wave of data collected (*n* = 389). The blue areas represent the boundaries of each village in the targeted area. The numbers and dots represent the number of questionnaires collected in each village.

**Figure 2 ijerph-19-09383-f002:**
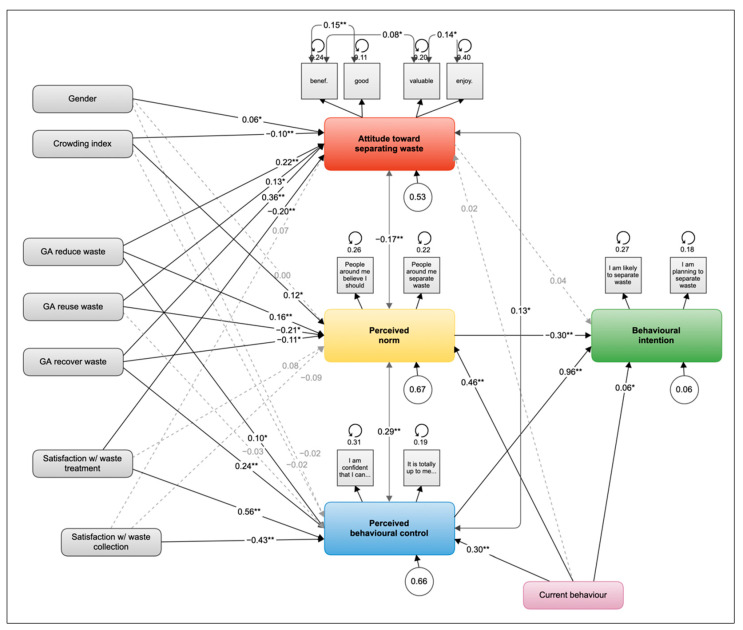
The tested TPB model with background factors. Notes: The latent factors representing the core TPB constructs are coloured; in grey are indicators and background factors; filled lines represent significant regression coefficients, dotted light grey arrows represent non-significant regression coefficients; numbers on paths represent the standardised path coefficients; asterisks represent the significance of the regression coefficient (* *p* < 0.05, ** *p* < 0.001); numbers in circles under each latent factor represent the unexplained variance (factor variances); and circles around indicators represent error variances (residual variances). For clarity, the path coefficients from background factors to intention are excluded except for current behaviour.

**Table 1 ijerph-19-09383-t001:** Characteristics of the total sample grouped by Regions.

Background Factors	Region 1 (*n* = 443)	Region 2 (*n* = 268)	Region 3 (*n* = 56)	Total (*n* = 767)	*p*-Value ^a^
Age, M (SD) {range}	45.9 (14.6) {18–85}	45.7 (14.6) {18–95}	49.3 (14.6) {20–80}	46.1 (14.6) {18–95}	0.202
Gender, *n* (%)					0.200
Female	252 (56.9)	166 (61.9)	31 (55.4)	449 (58.5)	
Male	191 (43.1)	102 (38.1)	25 (44.6)	318 (41.5)	
Education, *n* (%)					<0.001 **
No education	38 (8.7)	12 (4.5)	4 (6.7)	54 (7.0)	
Elementary	68 (15.5)	27 (10.1)	6 (10.0)	10 (13.1)	
Middle	91 (20.8)	47 (17.5)	13 (21.7)	151 (19.7)	
Secondary	105 (24.0)	52 (19.4)	14 (23.3)	171 (22.3)	
Vocational	46 (10.5)	25 (9.3)	3 (5.0)	74 (9.7)	
University	90 (20.5)	105 (39.2)	20 (33.3)	216 (28.1)	
Marital status, *n* (%)					0.065
Single	50 (11.4)	29 (10.9)	8 (13.6)	87 (11.4)	
Engaged/in a relationship	18 (4.1)	24 (9.1)	1 (1.7)	43 (5.6)	
Married	325 (74.0)	180 (67.9)	45 (76.3)	550 (72.1)	
Divorced/Separated	17 (3.9)	8 (3.0)	2 (3.4)	27 (3.5)	
Widowed	29 (6.6)	24 (9.1)	3 (5.1)	56 (7.3)	
Having children, *n* (%)	206 (47.0)	115 (42.9)	29 (48.3)	350 (45.7)	0.517
Crowding index, M (SD) {range}	1.0 (0.4) {0.0–2.7}	0.9 (0.6) {0.0–8.0}	1.2 (1.1) {0.0–8.0}	1.0 (0.6) {0.0–8.0}	0.007 **
Overcrowding, *n* (%)					<0.001 **
Acceptable crowding < 1 ppl/room	204 (46.5)	174 (64.7)	30 (50.0)	408 (53.1)	
Medium crowding (1–2 ppl/room)	230 (52.4)	90 (33.5)	25 (41.7)	345 (44.9)	
Overcrowding > 2 ppl/room	5 (1.1)	5 (1.9)	5 (8.3)	15 (2.0)	
General attitude toward waste, M (SD) {range}					
Reducing	8.7 (1.4) {3–10}	9.2 (1.1) {4–10}	9.2 (1.1) {5–10}	8.9 (1.3) {3–10}	<0.001 **
Reusing	8.0 (2.1) {1–10}	8.8 (1.4) {3–10}	8.7 (1.6) {4–10}	8.3 (1.9) {1–10}	<0.001 **
Recycling	8.3 (1.8) {1–10}	9.0 (1.4) {3–10}	9.0 (1.5) {4–10}	8.6 (1.7) {1–10}	<0.001 **
Recovering	8.4 (1.8) {2–10}	8.9 (1.5) {2–10}	9.1 (1.4) {4–10}	8.6 (1.7) {1–10}	<0.001 **
Satisfaction with current SWM system, M (SD) {range}					
Collection	6.1 (2.2) {1–10}	7.0 (2.2) {1–10}	5.0 (2.6) {1–9}	6.3 (2.3) {1–10}	<0.001 **
Waste treatment	4.7 (2.5) {1–10}	5.8 (2.7) {1–10}	4.6 (2.7) {1–9}	5.1 (2.6) {1–10}	<0.001 **
Behaviour: Do you separate waste?, *n* (%)					<0.001 **
Yes	96 (22.1)	94 (35.2)	6 (10.0)	196 (25.8)	
No	338 (77.0)	173 (64.6)	54 (90.0)	565 (73.7)	
TPB constructs, M (SD) {range}					
Attitude	8.5 (1.5) {4.33–10}	8.8 (1.3) {5–10}	8.9 (1.2) {5–10}	8.7 (1.4) {4.3–10}	0.003 *
Perceived social norm	4.7 (2.4) {1–10}	5.3 (2.7) {1–10}	4.2 (2.9) {1–9.5}	4.9 (2.9) {1–10}	0.001 *
Perceived behavioural control	6.6 (2.1) {1–10}	6.8 (1.7) {1–10}	6.7 (1.7) {1–10}	6.7 (1.9) {1–10}	0.379
Behavioural intention	7.0 (2.1) {1–10}	7.4 (1.7) {1.5–10}	7.5 (1.6) {2–10}	7.2 (1.9) {1–10}	0.020 *

Note: ^a^
*p*-value for ANOVA or Chi-square tests. * *p* < 0.05, ** *p* < 0.001; Region 1 = Union of municipalities of Iqlim el Tuffah; Region 2 = Union of municipalities of Jezzine; and Region 3 = Union of municipalities of Jabal al Rihan.

**Table 2 ijerph-19-09383-t002:** Internal consistency estimates, means, and factor loadings of the TPB constructs.

TPB Constructs	Cronbach’s Alpha (95% CI)	Mean (SD)	Factor Loadings	Residual Variance
Attitude toward waste segregation (Separating waste is…)	0.939 (0.920, 0.937)	8.642 (1.395)		
… beneficial (instrumental)		8.867 (1.382)	0.868	0.247
… good (instrumental.)		8.780 (1.464)	0.905	0.181
… valuable (instrumental)		8.654 (1.498)	0.934	0.128
… enjoyable (experiential)		8.266 (1.771)	0.794	0.37
Perceived norm (Most people around me…)	0.862 (0.841, 0.880)	4.866 (2.565)		
… believe I should separate waste (injunctive norm)		5.113 (2.794)	0.777	0.396
… separate waste (descriptive norm)		4.618 (2.676)	0.976	0.048
Perceived behavioural control	0.858 (0.836, 0.877)	6.689 (1.926)		
I am confident that I can separate waste (self-efficacy)		6.518 (2.052)	0.832	0.309
It is totally up to me whether I can separate waste (controllability)		6.860 (2.064)	0.903	0.184
Behavioural intention (In the coming month…)	0.872 (0.853, 0.889)	7.162 (1.938)		
… how likely are you to separate waste? (expectation)		6.978 (1.998)	0.91	0.172
… I am planning to separate waste (willingness)		7.346 (2.115)	0.852	0.275

## Data Availability

The dataset used in the path analyses and the full questionnaire are available from the corresponding author upon request.

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
