# Peer review of "Predicting the Intention to Sort Waste at Home in Rural Communities in Lebanon: An Application of the Theory of Planned Behaviour"

_ijerph, 2022, doi:10.3390/ijerph19159383_

Round 1
Reviewer 1 Report
The present study focused on the intention to sort waste at home in rural areas of Lebanon by using the theory of planned behavior method. This is quite interesting and important to waste management, particularly for low- and middle-income countries where the treatment methods are limited. The manuscript contains substantial information but it is more like a report than a scientific paper. Therefore, the quality of the manuscript needs to be significantly improved by making the introduction compact and logical, merging the sections, improving the resolution of the figures, and giving in-depth discussion by comparing other papers to provide useful indications for other LMICs. Please see my detailed comments as follows:
*Section 1-3, Introduction, Aims and Objectives of the Study, and Literature Review should be merged into one section to give the background, research updates, and research gaps of the present study.
*Why separately introduce “4. Theoretical and Research Framework”? Why not put it as the basis of the methods and materials?
*Fig.1: Please re-draw this figure since the quality is too hard to understand…
*Table 1: I suggest some key results should be presented as figures which can highlight the findings of the survey. Also for Tables 2 and 3.
Author Response
Response to Reviewer 1's comments (we upload also the complete response to the other reviewers' comments)
The present study focused on the intention to sort waste at home in rural areas of Lebanon by using the theory of planned behavior method. This is quite interesting and important to waste management, particularly for low- and middle-income countries where the treatment methods are limited. The manuscript contains substantial information but it is more like a report than a scientific paper. Therefore, the quality of the manuscript needs to be significantly improved by making the introduction compact and logical, merging the sections, improving the resolution of the figures, and giving in-depth discussion by comparing other papers to provide useful indications for other LMICs. Please see my detailed comments as follows:
Response 1.1. Thank you for your comments. We have amended the document to address your and the other reviewers’ suggestions to the best of our ability.
*Section 1-3, Introduction, Aims and Objectives of the Study, and Literature Review should be merged into one section to give the background, research updates, and research gaps of the present study.
*Why separately introduce “4. Theoretical and Research Framework”? Why not put it as the basis of the methods and materials?
Response 1.2. Thank you for your comments; we followed the structure of other manuscripts published in the same journal:
Labib, O. A., Manaf, L., Sharaai, A. H., & Zaid, S. S. M. (2021). Understanding the Effect of Internal and External Factors on Households’ Willingness to Sort Waste in Dammam City, Saudi Arabia. International Journal of Environmental Research and Public Health, 18(18), 9685. https://doi.org/10.3390/ijerph18189685
Liu, Q., Xu, Q., Shen, X., Chen, B., & Esfahani, S. S. (2022). The Mechanism of Household Waste Sorting Behaviour—A Study of Jiaxing, China. International Journal of Environmental Research and Public Health, 19(4), 2447. https://doi.org/10.3390/ijerph19042447
However, following your and other reviewers’ suggestions, we have shortened the introduction and merged the literature review with the theoretical framework into one paragraph. The methodology includes information about the TPB-based instrument we used, which we provide in the supplementary material.
*Fig.1: Please re-draw this figure since the quality is too hard to understand…
Response 1.3. We did not understand how the quality of the image was hard to understand.
We realised the picture was uploaded in a wrong way, so we re-uploaded a picture with a higher resolution, but we did not edit the picture as it represents the TPB and the text is clear.
*Table 1: I suggest some key results should be presented as figures which can highlight the findings of the survey. Also for Tables 2 and 3.
Response 1.4. Thank you for the suggestions, however, Table 1 represents the sample descriptives that cannot be summarised in figures. Table 2 also represents variable statistics and factor loadings that cannot be represented with figures. We removed Table 3 from the main article and kept the graph in Figure 3, as the graph summarises the information included in the table. We moved Table 3 in the appendix.

Reviewer 2 Report
Journal - IJERPH (ISSN 1660-4601)
Manuscript ID- ijerph-1827567
Manuscript: "Predicting the intention to sort waste at home in rural communities in Lebanon: An application of the Theory of Planned Behaviour"
Comments:
This study aimed to investigate the socio-cognitive and behavioral predictors of the 130 intention to sort waste at the source among citizens residing in the targeted area of the RES-Q project in Southern Lebanon. We focused on sorting waste at the source as it was considered a project priority. This manuscript should be major revised before considered for publication. There are a few grammatical and punctuation mistakes in the manuscript which need to be corrected. Besides, the following correction should be incorporated: -
General Comments.
1. The document, when analyzed on Plagiarisms software, i.e., Turnitin, is showing 17%. As per my view, it must be lower down up to 14% or so for an article. Self-Plagiarisms are also not accepted more than 2.5%.
2. The tables and figures used are not clear and can be enhanced. Heading must be with sequential numbers like 1.0, 1.1, 1.2, etc.
3. In reference section some reference is de-shaped, may be due to formatting. They are also needed to be corrected as per journal format.
4. The introduction must be reduced to one and a half pages.
5. The title needed significant modification.
6. The numbering of content must correct.
7. The manuscript requires an extension of the literature.
8. The manuscript does not illustrate great attention and activity in the field.
9. Tables also contain few references.
10. Please enhance the manuscript on analysis of earlier mention issues.
11. The figure number is distorted and can be rechecked.
Specific Comments.
1) Introduction: The authors should describe the importance of their research more clearly. The references cited lack articles on emerging contaminants from last year. So, add more references (2014-2021) to support the author's points of view. Abstract: It is needed to be started with small introduction and then quantitative description of the paper. It is also suggestive to shorten few unnecessary sentences in abstract.
3. Literature Review: Last paragraph must be an outline of the complete study showing the needed and targets assumed in the paper. Hence need minor revision. It also suggestive to add latest article in references.
4. Theoretical and Research Framework: More specific details are needed to be added with use of latest reference. It is suggestive to add grades of chemicals used and firm. Authors can also write step by step procedure. Figure 1 need more explanation in term of details w.r.t previous studies. A comparison table can be inserted for the same also.
5. Materials and Methods : More specific details are needed to be added with use of latest reference. RES-Q project details are needed to be discussed in tabular form. Discussion about the systesm used and its realtion needed to be defined well and justified before using it in the study.
5.2. Sampling and Procedures : More specific details are needed to be added with use of latest reference. Figure 2 quality need to be improved
7. Discussion: Cost analysis needed to be performed and industrial application needed to be discussed. Why and how the said parameters were selected for this experimentation. More specific details are needed to be added with use of latest reference. Use of some pictorial; diagram will be more elaborative for readers. Table 1 and Table 2,3 need more discussion in text.
Future scope of this study can be added as well as social impact can also be discussed in this paper.
Conclusions: This section is needed to be free from any variables are symbols. Only main pointed like what was expected and what was achieved must be written. What signification contribution this study to the society must be mentioned in this section.
Author Response
Response to Reviewer 2 Comments (we uploaded a document with the response to the other reviewers' comments)
Comments:
This study aimed to investigate the socio-cognitive and behavioral predictors of the intention to sort waste at the source among citizens residing in the targeted area of the RES-Q project in Southern Lebanon. We focused on sorting waste at the source as it was considered a project priority. This manuscript should be major revised before considered for publication. There are a few grammatical and punctuation mistakes in the manuscript which need to be corrected. Besides, the following correction should be incorporated: -
Response 2.1. Thank you for your comments. Based on other reviewers’ comments, we have completed major revisions of the introduction and of the rest of the manuscript. As we edited the document, we corrected any punctuation mistakes and grammar errors. The manuscript was reviewed by a native English speaker.
General Comments.
- The document, when analyzed on Plagiarisms software, i.e., Turnitin, is showing 17%. As per my view, it must be lower down up to 14% or so for an article. Self-Plagiarisms are also not accepted more than 2.5%.
Response 2.2. This is very weird as we checked the document for plagiarism and self-plagiarism using the plagiarism check on Word (see screenshot below). The six similarities included references in the reference list. As we heavily reviewed the manuscript, we checked the similarity index through Microsoft Word and Grammarly to ensure the writing is original.
- The tables and figures used are not clear and can be enhanced. Heading must be with sequential numbers like 1.0, 1.1, 1.2, etc.
Response 2.3. We have revised the quality of the images and uploaded ones with higher quality. The headings are consistent with the journal guidelines, as they were in the first submission.
- In reference section some reference is de-shaped, may be due to formatting. They are also needed to be corrected as per journal format.
Response 2.4. We are not sure what you meant by this comment. The references are entered using MDPI’s format, using the reference style files available from https://www.mdpi.com/journal/ijerph/instructions
- The introduction must be reduced to one and a half pages.
Response 2.5. Following reviewer #1’s comments, we revised the introduction and shortened it.
- The title needed significant modification.
Response 2.6. We are open to receiving suggestions about how to modify the title.
- The numbering of content must correct.
Response 2.7. We have checked that the headings are following a sequential numbering, as per journal guidelines.
- The manuscript requires an extension of the literature.
Response 2.8. The original manuscript included 80+ references, most of which related to the TPB applications; we believe the literature cited is sufficient, but we are open to suggestions about ways to include other literature that we might have missed.
- The manuscript does not illustrate great attention and activity in the field.
Response 2.9. We hope with the revised introduction and literature review sections, the manuscript provides now a better picture of the literature on solid waste management and behaviour change.
- Tables also contain few references.
Response 2.10. This comment is unclear, as we did not include any reference in our tables. We removed the reference included in Figure 1.
- Please enhance the manuscript on analysis of earlier mention issues.
Response 2.11. We have revised the entire manuscript to address the other reviewer’s comments hoping to address the issues.
- The figure number is distorted and can be rechecked.
Response 2.12. If the reviewer mentions figure 1, it was uploaded in the document specular by mistake. We have now included the correct image.
Specific Comments.
1) Introduction: The authors should describe the importance of their research more clearly. The references cited lack articles on emerging contaminants from last year. So, add more references (2014-2021) to support the author's points of view.
Response 2.13. We tried to find more recent articles, but there is a paucity of research from Lebanon on this topic; however, we tried to find more recent studies that describe the situation and the type of research done.
Abstract: It is needed to be started with small introduction and then quantitative description of the paper. It is also suggestive to shorten few unnecessary sentences in abstract.
Response 2.14. We revised the abstract to include some quantitative description of the paper noting the word limit. We believe all sentences are needed to explain what we did and how, but we are open to specific suggestions about unnecessary sentences.
- Literature Review: Last paragraph must be an outline of the complete study showing the needed and targets assumed in the paper. Hence need minor revision. It also suggestive to add latest article in references.
Response 2.15. As we mentioned in response 2.13, there is a paucity of research in Lebanon and the MENA region when it comes to waste management, but we tried our best to summarise the literature review with the most updated evidence.
- Theoretical and Research Framework: More specific details are needed to be added with use of latest reference. It is suggestive to add grades of chemicals used and firm. Authors can also write step by step procedure. Figure 1 need more explanation in term of details w.r.t previous studies. A comparison table can be inserted for the same also.
Response 2.16. We do not understand this comment as we did not include chemicals discussion in our paper. This comment does not seem to belong to our paper.
- Materials and Methods :More specific details are needed to be added with use of latest reference. RES-Q project details are needed to be discussed in tabular form. Discussion about the systesm used and its realtion needed to be defined well and justified before using it in the study.
Response 2.17. If we understand the comment, we provided more details about the RES-Q project, whose details have been published elsewhere. We did not discuss project details in tabular form as they were not relevant to the present study, which focused on the baseline data for the entire project. We included a few more details about the data collection methods used and the software we used to collect the data.
5.2. Sampling and Procedures : More specific details are needed to be added with use of latest reference. Figure 2 quality need to be improved
Response 2.18. If we understand the comment, we included more details about the sampling procedures that were used. We also uploaded a clearer Figure 2.
- Discussion: Cost analysis needed to be performed and industrial application needed to be discussed. Why and how the said parameters were selected for this experimentation. More specific details are needed to be added with use of latest reference. Use of some pictorial; diagram will be more elaborative for readers. Table 1 and Table 2,3 need more discussion in text.
Response 2.19. We do not understand this comment as it does not seem to belong to our paper. We have not mentioned industrial application or experimental applications.
Future scope of this study can be added as well as social impact can also be discussed in this paper.
Response 2.20. In the amended discussion, we added some more elaboration about the future scope of the study and the social impact.
Conclusions: This section is needed to be free from any variables are symbols. Only main pointed like what was expected and what was achieved must be written. What signification contribution this study to the society must be mentioned in this section.
Response 2.21. We did not include variables and symbols in the conclusions. If we understand this comment correctly, we revised the section to include a more elaborated part on the significance of the data and their implications for the future of the RES-Q project.
We included a paragraph on line 482 starting with: “How can these findings be used for developing a behaviour change strategy? First, interventions should focus on building behavioural skills linked to the PBC construct and critical mass to encourage the behaviour. Holding constant the positive level of attitude toward the behaviour, initiatives to address PBC should include building capability and the opportunity to perform the behaviour, following the COM-B model example [78]”.

Reviewer 3 Report
The study was conducted to determine the socio-cognitive and behavioral predictors of the intention to sort waste at the household among citizens residing in rural communities in Southern Lebanon. I am glad to evaluate the paper. Indeed, this is a good piece of research work. The Introduction establishes a good background to the research question. Materials and methods are well described. The analytical parameters are adequate for the interpretation of the data as well as for the conclusion. This is an original research, the findings of which has potential for increasing our understanding what motivates citizens to sort waste at home. This would help to identify of behaviour change and communication strategies to encourage recycling, reducing, and reusing waste at the household level. Therefore, I recommend this manuscript for publication in the Sustainability pending suggested revisions. My specific comments are given below:
L 220: Materials and Methods.
The description requires more detail of the references used in the data study. How representative sample of households was selected for the study? On what basis? Was the study guided by e.g.: of the socio-economic status ?, the number of people on the households or the number of rooms?. It would be worth clarifying how specific households were selected for the others.
L 230-232: 5.1. Questionnaire
I would suggest considering including the questionnaire as supplementary material.
L 318-44&
the chapter "Results" did not provide information on how to increase behavioral control citizens toward separating waste. It would be worthwhile to complete it, especially since the Abstract includes such examples.
L 545: Conclusions
Conclusions chapter is very general and seems to be a continuation of the Introduction chapter. Please edit it to summarize the most important results of this work.
Author Response
Response to Reviewer 3 Comments (we uploaded a document with the response to the reviewers' comments)
The study was conducted to determine the socio-cognitive and behavioral predictors of the intention to sort waste at the household among citizens residing in rural communities in Southern Lebanon. I am glad to evaluate the paper. Indeed, this is a good piece of research work. The Introduction establishes a good background to the research question. Materials and methods are well described. The analytical parameters are adequate for the interpretation of the data as well as for the conclusion. This is an original research, the findings of which has potential for increasing our understanding what motivates citizens to sort waste at home. This would help to identify of behaviour change and communication strategies to encourage recycling, reducing, and reusing waste at the household level. Therefore, I recommend this manuscript for publication in the Sustainability pending suggested revisions. My specific comments are given below:
Response 3. 1. Thank you for your appreciative and very constructive comments, which are very welcome. We have addressed them as described below.
L 220: Materials and Methods.
The description requires more detail of the references used in the data study. How representative sample of households was selected for the study? On what basis? Was the study guided by e.g.: of the socio-economic status ?, the number of people on the households or the number of rooms?. It would be worth clarifying how specific households were selected for the others.
Response 3.2. Thank you for this comment. We have added more details about the sampling procedure used based on the very limited data about the area (only the number of households). We added the details on lines 255-262, as follows:
“We used a random walk quota sample [68] stratified by the village to build a representative sample of the people living in the area. We generated a map of the area based on the dataset provided by the United Nations Office for the Coordination of Humanitarian Affairs (OCHA) [69]. We used ArcGIS [70] to generate a series of random points for each village; the weight for the random number generator was the estimated number of households in each village. The resulting layer of randomly generated points with coordinates was exported to Google Maps and imported in Kobotoolbox [71], the software used to collect data on tablets.”
L 230-232: 5.1. Questionnaire
I would suggest considering including the questionnaire as supplementary material.
Response 3.3. We have included a copy of the questionnaire used in the supplemental material.
L 318-44&
the chapter "Results" did not provide information on how to increase behavioral control citizens toward separating waste. It would be worthwhile to complete it, especially since the Abstract includes such examples.
Response 3.4. Thank you for pointing this out. The questionnaire did not ask about ways to increase behavioural control, so we cannot include data specific about this. However, it is a very important aspect of the study that deserves attention. Therefore, we expanded on the role of perceived behavioural control in the discussion, in which we included our interpretation of the data and ways to increase behavioural control.
We included a paragraph on line 482 starting with: “How can these findings be used for developing a behaviour change strategy? First, interventions should focus on building behavioural skills linked to the PBC construct and critical mass to encourage the behaviour. Holding constant the positive level of attitude toward the behaviour, initiatives to address PBC should include building capability and the opportunity to perform the behaviour, following the COM-B model example [78]”.
L 545: Conclusions
Conclusions chapter is very general and seems to be a continuation of the Introduction chapter. Please edit it to summarize the most important results of this work.
Response 3.5. We reviewed the conclusion paragraph to highlight the main learning points that our study provides.

Round 2
Reviewer 1 Report
The authors have addressed all my concerns.
Reviewer 2 Report
No more comments.